# Identification of Andrographolide as an Agonist of Bile Acid TGR5 Receptor in a Cell Line to Demonstrate the Reduction in Hyperglycemia in Type-1 Diabetic Rats

**DOI:** 10.3390/ph16101417

**Published:** 2023-10-05

**Authors:** Yingxiao Li, Kai-Chun Cheng, I-Min Liu, Juei-Tang Cheng

**Affiliations:** 1Department of Nursing, Tzu Chi University of Science and Technology, Hualien 970302, Taiwan; yxli@ems.tcust.edu.tw; 2Department of Pharmacy, College of Pharmacy, Tajen University, Pingtung 90741, Taiwan; kc-cheng@tajen.edu.tw (K.-C.C.); iml@tajen.edu.tw (I.-M.L.); 3Institute of Medical Sciences, Chang Jung Christian University, Tainan City 71101, Taiwan

**Keywords:** andrographolide, DPP-4, GLP-1, insulin, Takeda G-protein-coupled receptor, diabetic rats

## Abstract

Andrographolide (ADG) is contained in bitter plants, and its effects are widely thought to be associated with taste receptors. The current study used animal studies and cell lines to investigate the role of ADG in diabetic models. The Takeda G-protein-coupled receptor (TGR5) was directly influenced by ADG, and this boosted GLP-1 synthesis in CHO-K1 cells transfected with the TGR5 gene. However, this was not seen in TGR5-mutant cells. The human intestinal L-cell line NCI-H716 showed an increase in GLP-1 production in response to ADG. In NCI-H716 cells, the TGR5 inhibitor triamterene reduced the effects of ADG, including the rise in TGR5 mRNA levels that ADG caused. Additionally, as with the antihyperglycemic impact in type-1 diabetic rats, the increase in plasma-active GLP-1 level caused by ADG was enhanced by a DPP-4 inhibitor. The recovery of the hypoglycemic effect in diabetic rats and the increase in plasma GLP-1 caused by ADG were both suppressed by TGR5 blockers. As a result, after activating TGR5, ADG may boost GLP-1 synthesis in diabetic rats, enhancing glucose homeostasis. In Min-6 cells, a pancreatic cell line grown in culture, ADG-induced insulin secretion was also examined. Blocking GLP-1 receptors had little impact, suggesting that ADG directly affects TGR5 activity in Min-6 cells. A TGR5 mRNA level experiment in Min-6 cells further confirmed that TGR5 is activated by ADG. The current study revealed a novel finding suggesting that ADG may activate TGR5 in diabetic rats in a way that results in enhanced insulin and GLP-1 production, which may be helpful for future research and therapies.

## 1. Introduction

Andrographolide (ADG) is a labdane diterpenoid (Figure 1) found in the plant *Andrographis paniculata (Burm. F.) Wall. Ex.*, also known as Chun Xin Lián in China and sambiloto in India and Southeast Asia [1]. Sambiloto extract has recently been shown to raise plasma GLP-1 levels in prediabetic subjects [2]. Sambiloto is also known as “the king of bitters” due to its extremely bitter taste. This plant is used in Malaysian folk medicine to treat diabetes and hypertension [3]. Because of its bitter taste, it is known in Malaysia as “hempedu bumi”, which translates loosely as “earthly bile”. According to the pre-existing research on this issue, sambiloto’s bitter taste stimulates the release of GLP-1 [4]. As a result, ADG, a key ingredient in the sambiloto plant, may likewise be linked to GLP-1.

Recent research has discussed the various implications of ADG. ADG has been suggested as a new treatment option for metabolic syndrome [5]. Investigations found the impact of ADG on diabetic diseases and associated comorbidities to be substantial [6,7]. ADG was also observed to increase HDL-C levels in animals while decreasing plasma lipids [8]. Additionally, the use of ADG in animals provided benefits against obesity [9] and hypertension [10]. ADG has been demonstrated to boost the immune system [11,12] and lessen inflammation, possessing antibacterial and antiviral characteristics [13] in addition to its antihyperglycemic [14] and hepatoprotective effects [15]. ADG-induced anticancer activity has recently been demonstrated on number of cancer panels [16]. However, the primary mechanisms of ADG for treating diabetes are still not completely understood.

The objective of the current study is to use animal studies and cell lines to investigate the mechanisms responsible for ADG activity. Additionally, we conducted investigations into the relationship between ADG, GLP-1 and TGR5, which can influence insulin secretion and blood glucose levels.

**Figure 1 pharmaceuticals-16-01417-f001:**
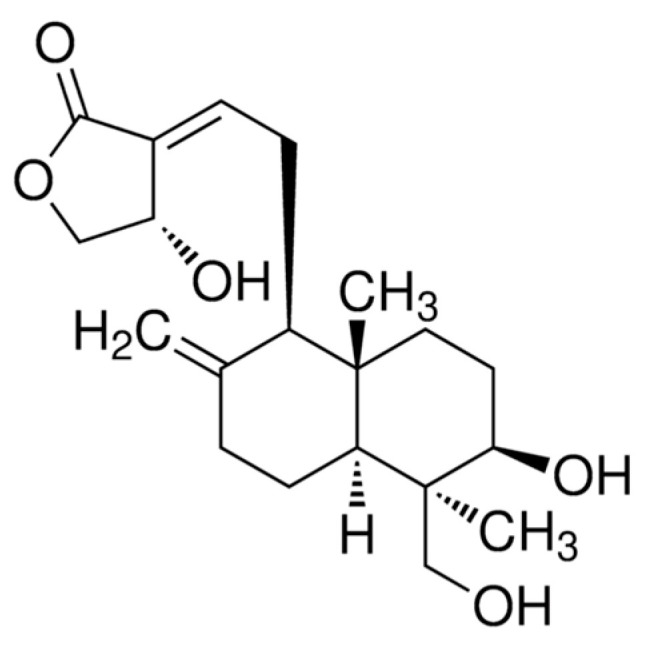
The chemical structure of andrographolide (ADG) [17].

## 2. Results

### 2.1. Preliminary Identification of Antihyperglycemic Effect of Andrographolide (ADG)

The role of GLP-1 in regulating the impact of ADG on fasting plasma glucose in STZ-treated rats has been investigated. Bolus injection of ADG reduced the hyperglycemia in STZ rats, and the effective dose of ADG was 1.5 mg/kg, as previously reported [18]. In order to inhibit DPP-4 activity, diabetic rats were given 5 mg/kg sitagliptin orally every day for two weeks prior to ADG treatment. As shown in Figure 2a, sitagliptin significantly increased the efficacy of ADG in lowering hyperglycemia. Additionally, the effect of ADG was abolished by exendin 9-39 (Ex-9) at the dose required to block the GLP-1 receptor. These findings suggest that GLP-1 may act as a mediator of ADG-induced hypoglycemia in humans. Then, we focused on the potential mechanism for GLP-1 secretion by ADG. Probenecid [19], like 6-methoxyflavanone (6-MeOF) [20], has been used as the antagonist of the bitter taste receptor (T2Rs). Probenecid, or 6-MeOF, could effectively block T2Rs when administered at 50 mg/kg (i.p.). However, neither probenecid nor 6-MeOF reduced ADG-induced hyperglycemia (Figure 2b), indicating that ADG does not appear to be associated with T2R activation. However, triamterene, which has been identified as a TGR5 receptor blocker [21], significantly reduced the effects of ADG. Thus, TGR5 appears to be crucial in the effects of ADG. 

### 2.2. Direct Effect of ADG on TGR5 Receptor In Vitro

We used CHO-K1 cells to transfect the exogenous TGR5 receptor gene, as described in our previous report [22]. The direct effect of ADG on the TGR5 receptor was then calculated using these cells. In TGR5-expressing cells, ADG significantly increased intracellular cAMP content as an indicator in a dose-dependent manner (Figure 3a). In CHO-K1 cells lacking TGR5 expression, ADG had no effect on cAMP content (Figure 3a). Furthermore, because the presence of the TGR5 receptor in NCI-H716 cells has been demonstrated [22], these cells are widely used in the study of GLP-1 secretion. Subsequently, we investigated how ADG affected GLP-1 secretion in vitro using NCI-H716 cells. ADG 182 increased GLP-1 secretion after incubation with NCI-H716 cells (Figure 3b). In these cells, triamterene inhibited ADG-induced GLP-1 secretion in a dose-dependent manner (Figure 3b). An mRNA level assay was used to investigate the effects of ADG on the TGR5 receptor in NCI-H716 cells.

As shown in Figure 3c, ADG increased TGR5 mRNA levels in a dose-dependent manner, and the same increase was observed in cells treated with oleanolic acid (OLA), another TGR5 agonist. Additionally, ADG increased intracellular calcium levels in NCI-H716 cells, as demonstrated in Figure 3d by the changes in GLP-1 secretion. Triamterene inhibited the ADG-induced increase in intracellular calcium levels at the same dose as TGR5 receptor blockade. ADG may collectively activate TGR5 in a way that increases cell GLP-1 secretion.

### 2.3. Effects of ADG on Plasma GLP-1 Levels in Type-1 Diabetic Rats

ADG increased plasma GLP-1 levels significantly in STZ-treated rats (Figure 4a). In these diabetic rats, pretreatment with sitagliptin at an effective dose in terms of inhibiting DPP-4, a GLP-1-inactivating enzyme, significantly increased this effect of ADG (Figure 3a). Furthermore, as shown in Figure 4b, ADG increased plasma beta-endorphin (BER) in these diabetic rats at the same dose that increased GLP-1 secretion. At the same dose, sitagliptin enhanced the effect of ADG. Surprisingly, it was also influenced by exendin 9-39 (Ex-9) and triamterene (Figure 4b). Therefore, ADG increased BER by inducing GLP-1 after the activation of TGR5 in these diabetic rats.

### 2.4. Effects of ADG on TGR5 to Induce Insulin Secretion from the Pancreatic Cell Line Min-6

Min-6 cells have been used to identify the direct effect of ADG on TGR5 in terms of insulin secretion in vitro. TGR5 is known to be expressed in pancreatic cells, and TGR5 has been shown to influence islet cell function [23]. As a result, we incubated ADG to examine the change in insulin secretion from Min-6 cells using OLA as a control. Similar to a previous report [23], ADG increased insulin secretion from Min-6 cells in a dose-dependent manner in the presence of 15 mmol/L glucose (Figure 5a). Triamterene inhibited this effect, but Ex-9 had no effect at the dose required to block the GLP-1 receptor (100 nmol/L). Notably, PKA inhibitors reduced ADG-induced insulin secretion (Figure 5b). In Min-6 cells, the same changes were observed in intracellular calcium levels (Figure 5c). TGR5 mRNA levels were determined in Min-6 cells via the use of OLA as a positive control, as shown in Figure 5d. TGR5 mRNA levels were raised by ADG in a dose-dependent manner. As a result, ADG may activate TGR5 in Min-6 cells in order to boost insulin secretion. 

To investigate the effects of ADG on pancreatic function in STZ-treated rats, two groups of diabetic rats were used in this in vivo study: one group received bolus injection of ADG (1.5 mg/kg) daily for one week, while the other group received a daily injection of vehicle at the same volume in parallel. Blood samples from these rats were used in the comparison. ADG reduced hyperglycemia from 358. 5 ± 9.8 mg/dL (n = 8) to 311.6 ± 6.9 mg/dL (n = 8), whereas it did change in the vehicle-treated group from 352. 8 ± 9.8 mg/dL (n = 8) to 348.8 ± 8.7 mg/dL (n = 8). Meanwhile, plasma biomarkers of pancreatic function, including plasma C-peptide and insulin levels, did not differ between the two groups. After one week of ADG treatment, plasma C-peptide levels in STZ-treated rats were 1.15 ± 0.11 ng/mL (n = 8) with no difference from previous treatments (1.22 ± 0.15 ng/mL; n = 8). The changes in the plasma C-peptide levels in the vehicle-treated group was the same, with a shift from 1.28 ± 0.21 ng/mL (n = 8) to 1.24 ± 0.15 ng/mL (n = 8). Furthermore, the same treatment with ADG had no effect on plasma insulin levels. Plasma insulin levels in STZ-treated rats were 1.05 ± 0.17 ng/mL (n = 8), with no difference from the control group (1.35 ± 0.22 ng/mL; n = 8). Plasma insulin levels in the vehicle-treated group changed by the same amount, shifting from 1.03 ± 0.15 ng/mL (n = 8) to 1.17 ± 0.14 ng/mL (n = 8). ADG treatment for one week had no effect on plasma biomarkers of pancreatic function. Acute ADG treatment appeared insufficient to the task of reversing the pancreatic damage caused by STZ in rats.

## 3. Discussion

In the current study, we discovered that ADG can induce GLP-1 secretion primarily through the activation of the bile acid TGR5 receptor. Furthermore, ADG had a direct effect on insulin secretion in a cultured pancreatic cell line (Min-6 cells) that expressed TGR5. ADG has been shown to be an agonist of the hTAS2R50 bitter taste receptor [24]. Therefore, the role of the bitter taste receptor and ADG in glycemic control in STZ-treated rats has been explored using the bitter taste receptor inhibitors probenecid [19] and 6-MeOF [25]. Probenecid was introduced as a complete and irreversible antagonist to the bitter taste receptor hTAS2R16 [19], whereas 6-MeOF [20] acts like MSG [25] as a surmountable antagonist. Both inhibitors had no effect on ADG-induced hypoglycemia. Bitter taste receptor mediation appears improbable. Triamterene is an inhibitor of the TGR5 receptor that has been used in a variety of experiments investigating MDMA-induced hyperthermia [21], Roux-en-Y gastric bypass (RYGB) [26], lung adenocarcinoma [27], and polycystic ovary syndrome (PCOS) via the microbiota–bile acid–IL22 axis [28]. Triamterene-induced IL-22 secretion was nearly identical to the level seen in TGR5-knockout mice [28]. As a result, animal studies have shown that triamterene is effective at inhibiting TGR5. The presence of triamterene at higher concentrations, on the other hand, has recently been shown to increase mycophagy in HepG2 cells unrelated to TGR5 [29]. Triamterene, like pharmacological inhibitors, must be used at the proper dose in order to inhibit TGR5. 

According to the current report, ADG has a direct effect on TGR5. ADG enhanced cellular calcium levels in CHO-K1 cells in a dose-dependent manner when the TGR5 gene was transfected into these cells. ADG, however, had no impact on calcium levels in CHO-K1 cells when TGR5 was absent. This study showed that ADG might regulate TGR5 activity, which could have a direct impact on GPCRs [23]. Furthermore, ADG may promote TGR5 activation by increasing TGR5 mRNA expression levels in cells, same as the well-known agonists [30]. TGR5 mRNA levels in pancreatic cells were significantly elevated as a result of ADG. ADG was also found to activate TGR5 in a manner that improved insulin secretion [23]. GLP-1 production from intestinal L-cells is stimulated by TGR5 activation [30]. In the current work, ADG was discovered to activate TGR5, and an increase in GLP-1 secretion was seen both in vivo and in vitro. As a result, our findings showed that ADG activates the TGR5 receptor which increases insulin production and GLP-1 levels. Many natural compounds, including betulinic acid [31] and glycyrrhizic acid [32], demonstrated effects on TGR5 activation in a manner similar to that of ADG. Recent docking tests with ADG and these medications may have revealed the binding location in the TGR5 homology model [33]. In order to develop novel chemicals in the future, it may be helpful to investigate the structure–activity relationship (SAR) between ADG and these molecules. 

Additionally, using ADG might increase the release of endogenous opioids, especially beta-endorphin (BER) [34], which is advantageous for glycemic control. ADG-induced plasma BER and GLP-1 release have been documented in the current study. A TGR5 receptor antagonist, triamterene, inhibited the increase in plasma BER and GLP-1 secretion brought on by ADG. That was comparable to a recent study [35] that suggested that BER may be released as a result of GLP-1 receptor activation. Therefore, more investigation is needed to ascertain whether ADG caused a BER-mediated antihyperglycemic action in STZ-diabetic rats.

TGR5 is expressed in pancreatic cells, and TGR5 agonists have been shown to increase insulin secretion [36]. Later, the same authors demonstrated that TGR5 agonists in the pancreas are associated with the GLP-1 released from intestinal L-cells, which acts on beta-cells in a paracrine manner [37]. TGR5 activation-induced insulin secretion appears to be dependent on both direct and indirect action via GLP-1. Ex-9, at the dose required to block the GLP-1 receptor, had no effect on ADG-induced insulin secretion in the current study. Our results confirm a recent study that ADG directly stimulates insulin production in Min-6 cells [23]. The main explanation for this would be the Min-6 cell line’s exclusive clonal beta-cell composition, which prevents GLP-1 released locally from other cell types from having an impact. In the present study, we have shown that ADG mediates GLP-1 release and enhances blood control in a diabetic model via activating signaling pathways that are stimulated by TGR5, as shown in Figure 6. It is essential to confirm this prior to using a type-2 diabetic model. The importance of ADG binding with TGR5 on intestinal L-cells or pancreatic islets in type-2 diabetes animals is another fascinating subject that needs more research. ADG has been proposed as a novel therapeutic strategy for the management of metabolic syndrome [5]. ADG has also been demonstrated to reduce blood sugar levels by modulating gut flora and enhancing intestinal barrier function [14]. The inhibition of indoleamine 2,3-dioxygenase by ADG [33] is unrelated to TGR5. As a result, ADG may activate the action of TGR5, which is crucial in combating metabolic syndrome. In the type-2 diabetes model, the priority of ADG binding with TGR5 on intestinal L-cells or pancreatic islets is also an intriguing topic that must be researched more in the future. 

## 4. Materials and Methods

### 4.1. Animals

Seven-week-old Sprague Dawley (SD) rats of a specific pathogen-free (SPF) male strain, weighing 240–260 g, were purchased from the National Laboratory Animal Center (Taipei, Taiwan). The animal experiments were approved by the Local Ethics Commission for Animal Experiments of Tajen University (IACUC 110-07). The animal experiments were conducted in accordance with the Guide for the Care and Use of Laboratory Animals of the National Institutes of Health, as well as the guidelines of the Animal Welfare Act.

### 4.2. Induction of Animal Model

Fasted rats received a single intravenous injection of STZ (65 mg/kg) to induce hyperglycemia, as previously described [38]. In this model, plasma insulin level was characterized as markedly reduced. After seven days of STZ treatment, rats were considered diabetic if their fasting plasma glucose level exceeded 300 mg/dL. Animal studies were then conducted one week after the onset of diabetes. Prior to the procedures, rats were anesthetized with a 35 mg/kg intraperitoneal injection of sodium pentobarbital in order to reduce animal suffering.

### 4.3. Fasting Plasma Glucose Levels Determined in Diabetic Rats Received the Testing Agents

Following the procedures of our previous method [18], ADG (98%) from Sigma-Aldrich (St. Louis, MA, USA) was dissolved in ethanol. After fasting overnight, diabetic rats were given a bolus injection of ADG at the indicated dose. They were then sedated with 2% isoflurane, and blood samples (0.1 mL) were drawn from the femoral vein to undergo plasma glucose measurement using a glucose kit with an automatic analyzer (Quik-Lab, Ames; Miles Inc., Elkhart, IN, USA). The same volume of vehicle (absolute alcohol in distilled water, 0.1 percent (*v*/*v*)) was given to the control group. Similarly, the inhibitor’s effect was determined after a 30 min pretreatment at the indicated dose. Otherwise, sitagliptin phosphate (Merck, Cramlington, Northumberland, UK) was diluted in normal saline as a DPP-4 inhibitor. Diabetic rats were then given 2 mg/kg/day sitagliptin orally for one week prior to ADG administration. Probenecid (Sigma-Aldrich), dissolved in 1 N NaOH and titrated to pH 7.0, was used to inhibit the bitter taste receptor. The same supplier provided another inhibitor, 6-MeOF. Triamterene (Sigma-Aldrich) has also been used as a TGR5 receptor blocker [21].

### 4.4. Cell Cultures

Cell lines were obtained from the Food Industry Institute’s Culture Collection and Research Center (Hsin-Chiu City, Taiwan). Human NCI-H716 cells (ATCC No. CCL-251) were cultured in RPMI 1640 supplemented with 10% (*v*/*v*) fetal bovine serum (FBS) and 2 mM L-glutamine in 5% CO_2_. F-12K growth medium containing 10% (*v*/*v*) FBS was also used to culture CHO-K1 cells (ATCC No. CCL-61). Min-6 cells were cultured in the same manner [39]. They were kept in an F-12K growth medium supplemented with 10% fetal bovine serum. Cells were sub-cultured using trypsin (GIBCO-BRL Life Technologies, Gaithersburg, MD, USA) every 3 days, and the medium was changed every 2–3 days.

### 4.5. TGR5 Transfection into CHO-K1 Cells

We transfected the human TGR5 gene into CHO-K1 cells using our previous method [22]. The successful transfection was confirmed the next day using the qPCR assay described below. TGR5-expressing cells were then treated for 1 h using ADG at the indicated concentrations. The intracellular cAMP levels were then determined using a commercial ELISA kit (Enzo Life Sciences, Farmingdale, NY, USA). Each assay was performed in duplicate for the indicated samples. The same treatment was performed on a control group of CHO-K1 cells that had not been transfected. The cAMP changes were then used as a reference. 

### 4.6. Determination of Intracellular Calcium Concentrations

Changes in intracellular calcium ([Ca^+2^]_i_) concentrations were detected using the fluorescent probe fura-2. To record the continuous fluorescence, a fluorescence spectrofluorometer was used (F-2000; Hitachi, Tokyo, Japan). The values of [Ca^+2^]_i_ were then determined. Background autofluorescence was measured in unloaded cells and subtracted from all measurements. The fluorescence values measured at 340 and 380 nm were used to calculate the [Ca^+2^]_i_ values. Furthermore, using a 30 min pretreatment, the effectiveness of the inhibitors, such as triamterene or others, was compared.

### 4.7. Assay of GLP-1 or Insulin Secretion from Cells

NCI-H716 cells were treated for 1 h with ADG at the indicated concentration. Some experiments included a 30 min incubation with triamterene prior to ADG treatment. After this, the supernatants were collected and tested for GLP-1 activity using a GLP-1 active ELISA kit (Millipore Co., Billerica, MA, USA). Each assay was performed in duplicate for the indicated samples. To determine the direct effect of ADG on insulin secretion in vitro [23], Min-6 cells were seeded in 12-well plates containing 1 mL of DMEM for 24 h prior to the experiment. 

A 30 min incubation with triamterene was a prerequisite in certain experiments to examine the role of TGR5 in ADG-induced insulin secretion. The Min-6 cells were also pretreated for 30 min with the inhibitor at the indicated concentrations or the vehicle control at the same volume in a KRBH buffer containing 15 mmol/L glucose. The cells were then incubated in ADG at the indicated concentration for 1 h. The supernatants were then collected and the insulin levels in the media were then determined using an insulin ELISA kit (Mercodia, Uppsala, Sweden) [40]. Additionally, each assay was carried out in duplicate.

### 4.8. Cells Real-Time Quantitative PCR

The mRNA levels of TGR5 were determined. A Trizol reagent was used to extract total RNA from cell lysates (Thermo Fisher, Carlsbad, CA, USA). Total RNA (200 ng) was reverse-transcribed into cDNA using random hexamer primers (Roche Diagnostics GmbH, Mannheim, Germany). PCR experiments were performed on a Light Cycler (Roche Diagnostics GmbH, Mannheim, Germany). A standard curve was used to calculate the concentration of each product. The target gene level was then divided by the level of β-actin in order to calculate relative gene expression. The following were the primers for each factor:TGR5 ReceptorF: 5′-TGGCTGCTGTGACTCTTTGA-3′R: 5′-TGTGACATCATGGGTCTTGG-3′β-actinF: 5′-CTAAGGCCAACCGTGAAAAG-3′R: 5′-GCCTGGATGGCTACGTACA-3′

### 4.9. Determination of Plasma GLP-1 Levels in Diabetic Rats

Diabetic rats were given 2 mg/kg/day sitagliptin (a DPP-4 inhibitor) or a vehicle orally for two weeks prior to ADG administration. Furthermore, the inhibitor’s effect was determined using a 30 min pretreatment of ADG at the indicated dose. Then, blood samples were taken from the femoral veins of the anesthetized rats. In a similar manner, plasma glucose levels were determined using an automated analyzer and a glucose kit (Quik-Lab, Ames; Miles Inc., Elkhart, IN, USA). Plasma-active GLP-1 levels were also measured using a commercial ELISA kit (EZGLP1T-36K, EMD Millipore Co., Billerica, MA, USA). Furthermore, plasma BER levels were determined via ELISA using a different commercial kit (Peninsula Laboratories, Belmont, CA, USA).

### 4.10. Statistical Analysis

The results are presented as the mean ± SEM of each group at the indicated sample size. They were analyzed using two-way analysis of variance on SPSS analysis software. Subsequently, they were subjected to Dunnett’s post hoc analysis (SPSS Inc., Chicago, IL, USA). A statistically significant *p* value of 0.05 was applied.

## 5. Conclusions

ADG can activate the bile acid TGR5 receptor both in vitro and in vivo, which has not previously been reported. Following TGR5 activation, ADG promotes GLP-1 secretion from intestinal L-cells to stimulate insulin secretion, and ADG may also induce insulin secretion directly from the pancreatic beta-cell line (Min-6 cells), where TGR5 is expressed. Therefore, ADG has the potential to be used as an anti-diabetic agent in the future.

## Figures and Tables

**Figure 2 pharmaceuticals-16-01417-f002:**
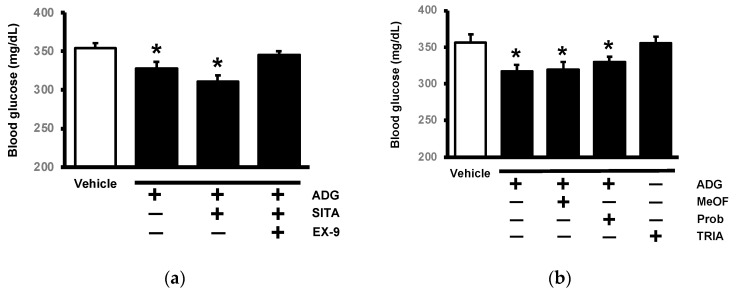
Characterization of the mediation of GLP-1 in the effects of andrographolide (ADG) on fasting plasma glucose (blood glucose) using STZ-induced diabetic rats. (**a**) The reduction in blood glucose via a bolus injection of ADG (1.5 mg/kg) was enhanced in diabetic rats given daily oral treatment with 5 mg/kg sitagliptin (SITA) for two weeks. This effect of ADG was, however, abolished by exendin 9-39 (Ex-9) at the intraperitoneal dose (0.1 mg/kg) required to block the GLP-1 receptor. (**b**) When compared to vehicle-treated diabetic groups (Vehicle), the reduction in blood glucose by ADG was not altered by 6-methoxyflavanone (MeOF) or probenecid (Prob) at the intraperitoneal dose (50 mg/kg) required to block the bitter taste receptor (T2Rs). Notably, it was significantly reduced by triamterene (TRIA) at an intraperitoneal dose (50 mg/kg) effective at blocking TGR5. Each group’s blood glucose value (mean ± SEM; n = 8) was displayed in a column. * *p* < 0.05 vs. vehicle-treated diabetic group.

**Figure 3 pharmaceuticals-16-01417-f003:**
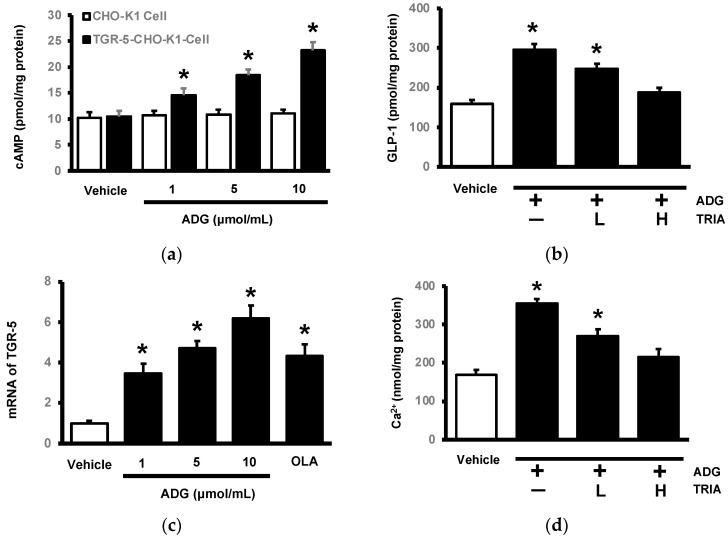
In vitro identification of andrographolide (ADG)-activated TGR5 in cells. (**a**) ANG induced cAMP elevation in CHO-K1 cells transfected with TGR5 (closed column) in a dose-dependent manner. It did not, however, have the same effect in CHO-K1 cells lacking TGR5 (open column). (**b**) ANG increased GLP-1 secretion in NCI-H716 cells, and pretreatment with triamterene (TRIA) may reduce this effect dose-dependently at low (L) and high doses (H). (**c**) ANG promoted TGR5 mRNA levels in NCI-H716 cells in a dose-dependent manner, and oleanolic acid (OLA), another TGR5 agonist, served as a reference. (**d**) ANG increased cellular calcium levels in NCI-H716 cells, while pretreatment with triamterene (TRIA) may also reduce this effect dose-dependently at low doses (L) and high doses (H). Each indicator’s value was displayed in a column as the mean ± SEM for each group (n = 8). * *p* < 0.05 compared to the vehicle-treated group.

**Figure 4 pharmaceuticals-16-01417-f004:**
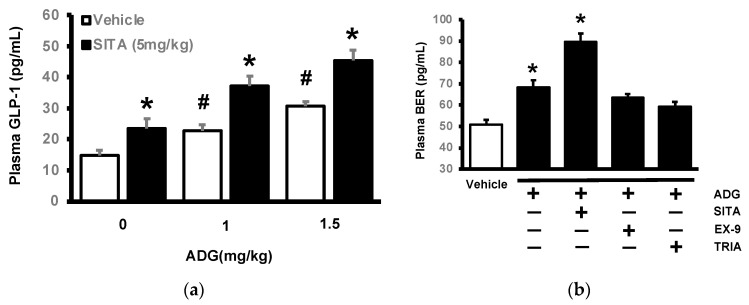
Identification of GLP-1 secretion caused by andrographolide (ADG) in diabetic rats. (**a**) ADG increased plasma GLP-1 levels dose-dependently (open column), and this effect was amplified in diabetic rats given daily oral treatment with 5 mg/kg sitagliptin (SI-TA) for two weeks (closed column). * *p* < 0.05 means the sitagliptin- + ADG-treated group (black column) compared to the diabetic group with or without ADG treatment (open column); ^#^*p* < 0.05 indicates the ADG-treated group (the second and third open columns) compared to the diabetic group receiving vehicle treatment (the first open column). (**b**) At the same dose, ADG also increased plasma beta-endorphin (BER) in these rats. It was enhanced in diabetic rats receiving SITA. Notably, changes in plasma BER caused by ADG were reduced by exendin 9-39 (Ex-9) at the dose (0.1 mg/kg, i.p.) required to block the GLP-1 receptor, as was TRIA (50 mg/kg, i.p.). Each indicator’s value is displayed in a column as the mean ± SEM for each group (n = 8). * *p* < 0.05 compared to the vehicle-treated group; # *p* < 0.05 shows the diabetic group treatment with ADG alone (open column) compared to the vehicle-treated group.

**Figure 5 pharmaceuticals-16-01417-f005:**
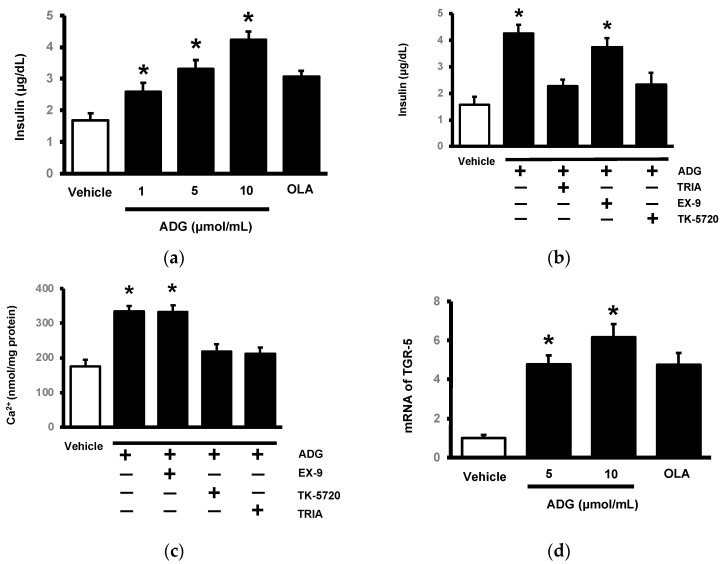
In vitro effect of andrographolide (ADG) on TGR5 in pancreatic cells. (**a**) Using oleanolic acid (OLA) as a control, ADG increased insulin secretion from Min-6 cells in a dose-dependent manner. (**b**) Triamterene (TRIA) inhibited ADG-induced insulin release but not exendin 9-39 (Ex-9). Furthermore, the PKA inhibitor KT5720 inhibited ADG’s action. (**c**) ADG increased calcium levels in Min-6 cells in the same way. ADG’s effect was mitigated by TRIA and KT5720, but not by Ex-9. (**d**) Using OLA as a positive control, cells were treated with ADG at the indicated concentration for 1 h. ADG promoted TGR5 mRNA levels from low doses to high doses in a dose-dependent manner. Each indicator’s value is displayed in a column as the mean ± SEM for each group (n = 8). * *p* < 0.05 compared to the vehicle-treated group.

**Figure 6 pharmaceuticals-16-01417-f006:**
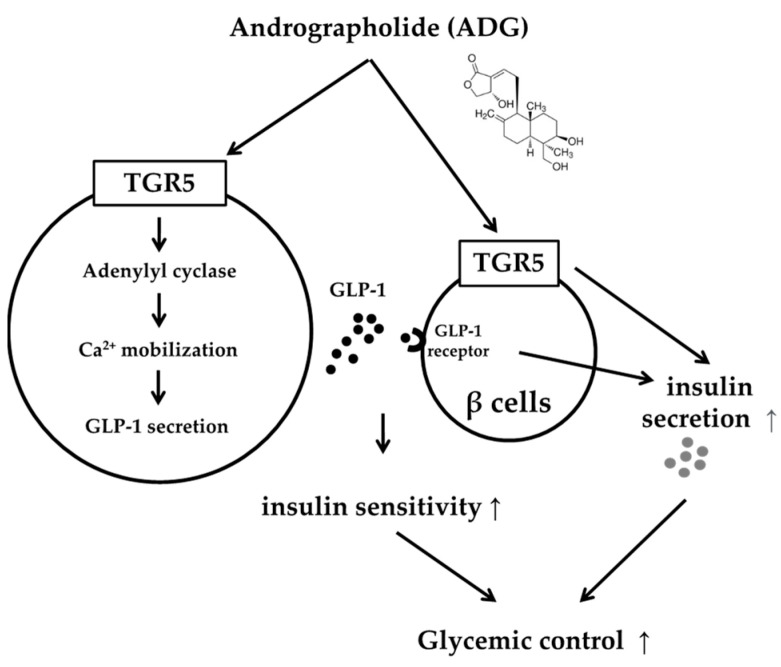
The action mechanisms of andrographolide in glucose homeostasis.

## Data Availability

Data is contained within the article. The correspondent author can provide the data included in the present study upon reasonable request.

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
