# Peer review of "Identification of Andrographolide as an Agonist of Bile Acid TGR5 Receptor in a Cell Line to Demonstrate the Reduction in Hyperglycemia in Type-1 Diabetic Rats"

_pharmaceuticals, 2023, doi:10.3390/ph16101417_

Round 1

Reviewer 1 Report

Manuscript: Identification of Andrographolide as Agonist of Bile Acid TGR5

Receptor in cell line to demonstrate the reduction of hyperglycemia in type-1 diabetic rats

General comments:
In the manuscript, the authors investigated the reduction effects of andrographolide (ADG) on hyperglycemia through the agonist of bile acid TGR5 receptor. In general, the authors have completed a reasonable study with very informative data on the relationships of the ADG effects and their potential activity mechanism. The statistical analysis and graphic presentation also have been completed in details. However, the presentation of this study may be strengthened by the following specific comments.

1. Andrographolide (ADG)
Q. Please present a figure of chemical structure of ADG.

2. Line 49, Sprague Dawley (SD) rats
Q. The years-old of rats should be presented in the study.

3. Line 98,  intracellular calcium (((Ca2+))i)
Please rephrase (((Ca2+))i) to [Ca+2]i

4. Figure 2. I In vitro identification of andrographolide
Please rephrase the sentence.

5. Figure 4, ADG promoted TGR5 mRNA levels
Please describe the induction time of ADG treatment in the legend before the conduction of mRNA studies.

6. Line 300, when using ADG, it is critical to ensure that TGR5 is turned on in the main.
Q. Please rephrase the ambiguous sentence. What is the “in the main”?

English editing is fine, minor edition is requires. 

Author Response

General comments:
In the manuscript, the authors investigated the reduction effects of andrographolide (ADG) on hyperglycemia through the agonist of bile acid TGR5 receptor. In general, the authors have completed a reasonable study with very informative data on the relationships of the ADG effects and their potential activity mechanism. The statistical analysis and graphic presentation also have been completed in details. However, the presentation of this study may be strengthened by the following specific comments.

1. Andrographolide (ADG)
Q. Please present a figure of chemical structure of ADG.
Reply: The figure of chemical structure of ADG has been provided. Thank you very much.

  1. Line 49, Sprague Dawley (SD) rats
    Q. The years-old of rats should be presented in the study.
    Reply: The years-old of rats has been included in the Methods section. Thank you very much.
  2. Line 98, intracellular calcium (((Ca2+))i)
  3. Please rephrase (((Ca2+))i) to [Ca+2]i

Reply: We have improved this part according to your comment. Thank you very much.

  1. Figure 2. I In vitro identification of andrographolide
    Please rephrase the sentence.

 Reply: We apologize for this oversight, and the text has been revised accordingly. Thank you.

  1. Figure 4, ADG promoted TGR5 mRNA levels
    Please describe the induction time of ADG treatment in the legend before the conduction of mRNA studies.

Reply: The induction time of ADG treatment has been added in the legend of Figure 4. Thank you very much.

  1. Line 300, when using ADG, it is critical to ensure that TGR5 is turned on in the main.
    Q. Please rephrase the ambiguous sentence. What is the “in the main”?

Reply: Thank you for this comment. We have improved the Discussion section according to your comment.

Certification of Editing:

Reviewer 2 Report

Dear Authors

here below i present my detailed comments to your interesting work:

- in the introduction, please correct the initials of the plant. We use italics for the name of the species, but the initials of the finders should be written with normal letters. please, introduce a suitable correction

- also, the name of the species is 'paniculata' instead of 'paniculate'

- the aim of the study is not clearly formed in the introduction. please, work on that. at the moment it is chaotic. we can read about the plant, its names, properties, and then again we are coming back to the characteristics of the plant  that is used in India and again about its properties. please, order these information, state clearly what is known and which information are still missing and which questions are going to be answered in this manuscript

- line 114 - have the Authors used imperatorin? please, comment on that

- please, provide a figure to your work that will explain the mechanisms of action that your study is targeting. please, add a little bit more explanation to your work on the meaning of the studied factors. Your work is very concise and relates to your results, but it would be much more interesting to read a little bit about the background and significance of the parameters that you are studying.

other minor comments

-line 158, please add a fullstop before '. Bolus'

- please, go through your work again and change in vitro to italics

- please, adjust the references style to the demandings of the journal

Please, be so kind to read abstract and introduction again

Author Response

- in the introduction, please correct the initials of the plant. We use italics for the name of the species, but the initials of the finders should be written with normal letters. please, introduce a suitable correction

Reply: As suggested, we have amended the he initials of the plant in Introduction section. Thank you very much.

- also, the name of the species is 'paniculata' instead of 'paniculate'

Reply: We have improved this part according to your comment. Thank you very much.

- the aim of the study is not clearly formed in the introduction. please, work on that. at the moment it is chaotic. we can read about the plant, its names, properties, and then again, we are coming back to the characteristics of the plant that is used in India and again about its properties. please, order this information, state clearly what is known and which information are still missing and which questions are going to be answered in this manuscript

Reply: Thank you for the helpful comments. We have provided a better explanation in the Introduction section. Thank you very much.

- line 114 - have the Authors used imperatorin? please, comment on that

Reply: As suggested, we have amended the Methods section. Thank you very much.

- please, provide a figure to your work that will explain the mechanisms of action that your study is targeting. please, add a little bit more explanation to your work on the meaning of the studied factors. Your work is very concise and relates to your results, but it would be much more interesting to read a little bit about the background and significance of the parameters that you are studying.

 Reply: We have improved the Discussion section in the revised version of the manuscript. Thank you very much.

other minor comments

-line 158, please add a full stop before '. Bolus'

Reply: Thank you for the helpful suggestion. The text has been revised accordingly.

- please, go through your work again and change in vitro to italics

 Reply: We have revised this part accordingly. Thank you very much.

- please, adjust the references style to the demanding of the journal.

 Reply: We have revised the references style following the instructions. Thank you very much.

Comments on the Quality of English Language

Reply: We have improved the Quality of English Language as suggested. Please find the certificate below. Thank you very much

Please, be so kind to read abstract and introduction again

Reply: Thank you for the helpful indications. We have improved the Introduction section as suggested.

Certification of Editing:

Round 2

Reviewer 2 Report

Dear Authors, in the author's reply section I found this fragment:

"- please, provide a figure to your work that will explain the mechanisms of action that your study is targeting. please, add a little bit more explanation to your work on the meaning of the studied factors. Your work is very concise and relates to your results, but it would be much more interesting to read a little bit about the background and significance of the parameters that you are studying.

Reply: We have improved the Discussion section in the revised version of the manuscript. Thank you very much."

however i do not find the requested graph or figure...

Author Response

Reply:We apologized for the omission of the question. Following your instructions, we added the mechanism of andrographolide affects glucose homeostasis in Figure 5, which also has been described in the Discussion section. Hope to have your kind agreements. Thank you very much.
